# Technology Invention and Mechanism Analysis of Rapid Rooting of *Taxus × media* Rehder Branches Induced by *Agrobacterium rhizogenes*

**DOI:** 10.3390/ijms25010375

**Published:** 2023-12-27

**Authors:** Ying Wang, Xiumei Luo, Haotian Su, Ge Guan, Shuang Liu, Maozhi Ren

**Affiliations:** 1Functional Plant Cultivation and Application Innovation Team, Institute of Urban Agriculture, Chinese Academy of Agricultural Sciences, Chengdu 610230, China; wangying08@caas.cn (Y.W.); guange0106@163.com (G.G.); liushuang061699@sina.com (S.L.); 2Hainan Yazhou Bay Seed Laboratory, Sanya 572025, China; 3School of Agricultural Science, Zhengzhou University, Zhengzhou 450052, China

**Keywords:** *Taxus × media* Rehder, *Agrobacterium rhizogenes*, branch rooting, paclitaxel

## Abstract

*Taxus*, a vital source of the anticancer drug paclitaxel, grapples with a pronounced supply–demand gap. Current efforts to alleviate the paclitaxel shortage involve expanding *Taxus* cultivation through cutting propagation. However, traditional cutting propagation of *Taxus* is difficult to root and time-consuming. Obtaining the roots with high paclitaxel content will cause tree death and resource destruction, which is not conducive to the development of the *Taxus* industry. To address this, establishing rapid and efficient stem rooting systems emerges as a key solution for *Taxus* propagation, facilitating direct and continuous root utilization. In this study, *Agrobacterium rhizogenes* were induced in the 1–3-year-old branches of *Taxus* × *media* Rehder, which has the highest paclitaxel content. The research delves into the rooting efficiency induced by different *A. rhizogenes* strains, with MSU440 and C58 exhibiting superior effects. Transcriptome and metabolome analyses revealed *A. rhizogenes*’ impact on hormone signal transduction, amino acid metabolism, zeatin synthesis, and secondary metabolite synthesis pathways in roots. LC-MS-targeted quantitative detection showed no significant difference in paclitaxel and baccatin III content between naturally formed and induced roots. These findings underpin the theoretical framework for *T. media* rapid propagation, contributing to the sustainable advancement of the *Taxus* industry.

## 1. Introduction

*Taxus* is a nationally protected plant in China that integrates medicinal, health, ornamental, and ecological values. It is regarded as a “national treasure” by countries around the world. Paclitaxel, an extract from *Taxus* trees, has broad-spectrum and highly effective anti-cancer activity. It is one of the most effective natural anti-cancer drugs currently available. After extensive mechanistic and clinical research, paclitaxel has been applied in the treatment of various cancers such as advanced ovarian, breast, and lung cancer. It is internationally recognized as a star anti-cancer drug [1,2]. Currently, the global demand for paclitaxel exceeds 3000 kg per year but the actual annual production is only about 300–500 kg. The domestic demand in China is over 500 kg per year but the actual production of paclitaxel is only around 200 kg. The supply and demand of paclitaxel, both internationally and domestically, are severely imbalanced.

*Taxus × media* Rehder has the highest content of paclitaxel among all *Taxus* species worldwide. Leaves and branches contain paclitaxel at concentrations of 0.015% and 0.018%, respectively, whereas the roots and bark contain as much as 0.071% paclitaxel [3]. It is estimated that the paclitaxel extracted from the bark of 3–6 approximately 100-year-old *Taxus* trees is only sufficient to meet the needs of treating one cancer patient. The long growth cycle and limited resources of wild *Taxus* trees make them unable to meet the demand for paclitaxel production [4]. Moreover, in the pursuit of paclitaxel for anti-cancer purposes, wild *Taxus* trees have been extensively logged, resulting in a severe depletion of this precious resource. Currently, cutting propagation is the main method for the artificial breeding of *Taxus* trees. It involves cutting branches from the trees for rooting but this method faces challenges such as low rooting success, low survival rates, and long requirements, which are not conducive to large-scale industrial development for paclitaxel extraction and other purposes.

The artificial propagation of *Taxus* tree seedlings is difficult and the supply–demand imbalance of paclitaxel has not been effectively alleviated. The main reason for this lies in the underutilization of the roots, which contain the highest concentration of paclitaxel. Accelerating the breeding of *Taxus* tree seedlings and overcoming the rooting bottleneck to increase the paclitaxel content remains a key focus for development support in countries worldwide. *Agrobacterium rhizogenes* is a soil bacterium with a wide range of infectivity. It can infect almost all dicotyledonous plants, inducing the plants to produce a large number of highly branched adventitious roots, eliminating some monocotyledonous plants [3]. The adventitious roots produced by the infection of *A. rhizogenes* have advantages such as a fast growth rate, high degree of differentiation; physiological, biochemical, and genetic stability; and ease of manipulation and control [5]. By relying on *A. rhizogenes*, inducing the aerial rooting of *T. media* branches (air-layering) not only shortens the artificial breeding cycle but also enables the sustainable and efficient utilization of the roots with a higher paclitaxel content. This approach will effectively enhance the economic value of *T. media*, increase paclitaxel production, and promote the development of the *Taxus* industry.

## 2. Results

### 2.1. High-Efficiency Induction of T. media Branch Rooting Using A. rhizogenes

After approximately seven months of induction treatment, 1–3-year-old branches of *T. media* successfully formed adventitious roots (Figure 1). In comparison to non-rooting branches, the rooting branches remained alive. After cleaning the substrate, it was observed that the roots grew from the girdling site, exhibiting dense and robust root growth ranging from 3 to 8 cm in length (Figure 2a,b). It is worth noting that the tips of the rooting branches were green and alive and the roots could be seen clearly through the high-pressure propagation box (Figure 1). In contrast, none of the branches survived root successfully. The rooting rate (proportion of rooted branches to total experimental branches) of non-treated *T. media* branches was only 14.43%. However, when treated with *A. rhizogenes* strains Qual, MSU440, Ar1193, C58, ATCC15834, and A4, the rooting rates were significantly increased to 50.00%, 55.23%, 40.78%, 55.00%, 48.45%, and 37.63%, respectively (Figure 2c). Meanwhile, untreated branches produced fewer adventitious roots, only 40 ± 13 (*n* = 10) in total. In contrast, branches treated with *A. rhizogenes* generated a higher number of adventitious roots, with the MSU440 treatment showing the most significant effect (Figure 2d). The experimental results demonstrated that compared to the control, all different *A. rhizogenes* treatments significantly improved the rooting rate of *T. media* branches, with MSU440 and C58 treatments showing better effects.

The rooting branches induced by *A. rhizogenes* strains MSU440 and C58 were cut 5 cm below the girdling site and transplanted into pots. They were managed according to the regular field management practices for *T. media*. All transplanted branches survived. The plants exhibited healthy growth and robust development and the growth status of the branches after one month of cultivation in the pots is shown in Figure 2d.

### 2.2. Analysis of the Gene Expression Profile in TR

In order to further elucidate the roles of *A. rhizogenes* in the induction of root formation in the branches of *T. media*, the RNA-seq of roots induced by MSU440 (TR) was performed. The RNA-seq of roots that grow naturally (WR) was taken as a control. After filtering the raw data that contained an adapter or was low-quality, clean reads were obtained for genome https://www.ncbi.nlm.nih.gov/genome/81451 (accessed on 10 May 2023) mapping and subsequent analysis. In summary, over 87% of the clean reads could be mapped to the annotated genome of *T. chinensis* (Appendix A); the correlation coefficient between samples is over 0.8, indicating that the similarity of expression patterns between samples was incredible (Appendix A). Compared with WR, a total of 7750 differentially expressed genes (differential expression analysis, DEGs) were found in TR, of which 4183 DEGs were upregulated and 3567 DEGs were downregulated (Figure 3a; Appendix A).

### 2.3. Gene Ontology (GO) and Kyoto Encyclopedia of Genes and Genomes (KEGG) Pathway Enrichment Analysis of DEGs

To illustrate the function of DEGs, GO functional enrichment was performed, including with respect to the biological process, cellular component, and molecular function (Figure 3b). A total of 2915 upregulated GO terms and 2843 downregulated GO terms were enriched in the RNA-seq data (Appendix A). Among the upregulated GO terms, the top two significantly enriched GO terms in biological process were the oxidation–reduction process and response to stimulus; those in the cellular component were chromosome and chromatin; and those in the terms of the molecular function were oxidoreductase activity and heme binding (Appendix A). Among the downregulated GO terms, the top two significantly enriched GO terms in the biological process were the biological process and metabolic process; those in the cellular component were the membrane and intrinsic (integral) components of the membrane; and those in terms of the molecular function were the catalytic activity and transferase activity (Appendix A).

The KEGG pathway enrichment analysis showed that metabolic pathways, the biosynthesis of secondary metabolites, and phenylpropanoid biosynthesis were the top three significant enrichment pathways, which were also enriched in downregulated DEGs (Figure 3c and Appendix A); among upregulated DEGs, the most significantly enriched three KEGG pathways were the biosynthesis of secondary metabolites, phenylpropanoid biosynthesis, and glutathione metabolism (Appendix A). These results indicate that *A. rhizogenes* mainly regulates metabolic processes in root formation.

### 2.4. Analysis of Differentially Expressed Metabolites in TR

To explain the chemical basis of metabolite accumulation in the roots, a metabolomics analysis was conducted using UHPLC-MS-MS technology. A total of 656 metabolites was detected in the six samples from the two treatments, among which 128 flavonoids and 67 terpenoids were the top two enriched metabolites (Appendix A). In addition, 58 alkaloids, 48 phenol and phenol ethers, 40 amino acids and their derivatives, 25 phenylpropanoids, and 20 nucleotides and their derivatives were also identified in the roots, indicating that they are rich in metabolites that can provide medicinal value (Appendix A). It is worth noting that 7 phytohormones including (+)-abscisic acid, 1-Aminocyclopropanecarboxylic acid, Indole-3-carboxaldehyde, DL-Dihydrozeatin, cis-Zeatin, N6-isopentenyladenosine, and trans-Zeatin-riboside were detected in the samples (Appendix A). Studies have shown that secondary metabolites such as paclitaxel and baccatin III were detected in the roots of TR and WR, which further indicated that the newly grown roots also can synthesize paclitaxel (Appendix A).

Based on the screening criteria (*p*-value < 0.05 and VIP ≥ 1) for identifying differential metabolites, 229 DEMs were identified in the TR vs. WR. Among them, 93 DEMs were upregulated and 136 DEMs were downregulated (Figure 4a). Principal component analysis revealed clear differences between the TR and WR (Figure 4b). K-means clustering analysis suggested that DEMs might be subdivided into six levels according to the trend in content change in each treatment (Figure 4c and Appendix A). Further DEMs analysis indicated that most of the secondary metabolites such as flavonoids (48/58), terpenoids (14/15), and alkaloids (11/16) were downregulated, while most of the metabolites related with growth as amino acids and their derivatives (21/22) and nucleotides and their derivatives (10/12) were upregulated (Appendix A). Subsequent KEGG enrichment analysis showed that these DEMs were the main components for the biosynthesis of amino acids and their derivatives (Appendix A). Furthermore, the trans-zeatin-riboside, which plays an important role in cell division, was upregulated in TR vs. WR (Appendix A).

### 2.5. Combined Analysis of Transcriptome and Metabolome (DEGs/DEMs) in TR vs. WR

A total of 67 KEGG pathways correlated with DEGs/DEMs were obtained in comparisons of TR vs. WR, including 1972 genes and 252 metabolites (Appendix A). The pathways are mainly divided into three categories: carbon metabolism (11 KEGGs, 326 DEGs, and 20 DEMs), biosynthesis of amino acids (14 KEGGs, 198 DEGs, and 48 DEMs), and biosynthesis of secondary metabolites (9 KEGGs, 525 DEGs, and 43 DEMs). Comprehensive analysis of KEGG pathways and −log(*p*-value) and correlated DEGs/DEMs were mainly involved in plant hormone signal transduction (ko04075), amino acid (alanine, aspartate, and glutamate) metabolism (ko00250), and zeatin biosynthesis (ko00908), while a large number of DEGs/DEMs was involved in the biosynthesis of secondary metabolites (ko01110) (Figure 5a; Appendix A).

### 2.6. DEGs/DEMs Were Involved in Plant Hormone Signal Transduction

A crucial aspect of plant hormone function involves the regulation of diverse developmental processes. Hormones such as auxin, jasmonic acid (JA), gibberellic acid (GA), cytokinin, abscisic acid (ABA), ethylene, brassinosteroid, salicylic acid (SA), and strigolactone (SL) employ mechanisms that activate transcriptional regulators for degradation. As shown in Figure 5b and Appendix A, in the auxin signaling pathway, DEGs encoding Aux1 were consistently downregulated, while those for TIR1, AUX/IAA, and ARF were consistently upregulated. Cytokinin, known for its role in cell proliferation, exhibited upregulation of core DEGs, including CRE1 and AHP. ABA, a critical endogenous messenger in plant stress responses, showed upregulation of DEGs encoding ABA receptor genes PYR/PYL but downregulation of those encoding the pivotal regulator SnRK2. In the ethylene signaling pathway, DEGs encoding protein kinase CTR1 and F-Box proteins EBF1/2 were upregulated. Moreover, genes (*JAR1*, *COI1*, *NPR1*, and *TGA*) involved in JA and SA signal transduction pathways were upregulated.

### 2.7. DEGs/DEMs Were Involved in Amino Acid Metabolism

In plants, amino acids serve multiple functions associated with growth. Besides their function in protein synthesis, the amino acids are also catabolized into energy-associated metabolites as well as into numerous secondary metabolites, which are essential for plant growth and response to various stresses [6]. Apart from alanine, aspartate, and glutamate metabolism (ko00250); tyrosine metabolism (ko00350); phenylalanine metabolism (ko00360); glycine, serine, and threonine metabolism (ko00260); histidine metabolism (ko00340); lysine biosynthesis (ko00300); arginine and proline metabolism (ko00330); cysteine and methionine metabolism (ko00270), valine, leucine, and isoleucine degradation (ko00280); lysine degradation (ko00310); phenylalanine, tyrosine, and tryptophan biosynthesis (ko00400); tryptophan metabolism (ko00380); valine, leucine, and isoleucine biosynthesis (ko00290); and arginine biosynthesis (ko00220) were also enriched in TR vs. WR. DEMs analysis showed that 95% of amino acids were upregulated and most DEGs related to amino acid metabolism were also upregulated (Appendix A).

### 2.8. DEGs/DEMs Were Involved in Zeatin Biosynthesis

Cytokinin hormones are important regulators of the development and environmental responses of plants that execute their action via the molecular machinery of signal perception and transduction [7]. Trans-zeatin and isopentenyl adenine-type cytokinins were thought to be the predominant cytokinins, with the cisisomer being present only in minor quantities with low or no activity. There are a growing number of reports on cis-zeatin being the dominant cytokinin species in various plants such as potatoes. As shown in Figure 5c and Appendix A, in the zeatin biosynthesis pathway (ko00908), eight genes encoding CYP735A, UGT73C1, and enzymes catalyzing isopentenyl-adenine were differentially expressed, which were almost up-regulated. The metabolites DL-Dihydrozeatin and trans-zeatin-riboside were downregulated by 1.0674-fold and upregulated by 1.303-fold, respectively. 

### 2.9. The Paclitaxel and Baccatin III Were Synthesized in TR and WR

To determine the content of paclitaxel and baccatin III in TR and WR, quantitative detection was performed. The contents of paclitaxel and baccatin III were also detected in the 20-year-old roots. The standard curve of paclitaxel is y = 1056.5x + 11338, with an R^2^ is 0.9981; and that of baccatin III is y = 437.9x + 6054.6, with an R^2^ is 0.9994 (Appendix A). The results of quantitative detection showed that the paclitaxel content was 321 ng/mg in TR and 305 ng/mg in WR, respectively, which was not significantly different (Figure 6a,b). In the 20-year-old roots, the paclitaxel content was 5792 ng/mg, which was almost 18 times that in TR. For baccatin III, the content in TR was 0.63 ng/mg and that in WR was 0.62 ng/mg. While in the 20-year-old roots, the baccatin III content was 2.72 ng/mg, which was almost 4.2 times that in TR (Figure 6a,c). The results indicated that paclitaxel and baccatin III can also be synthesized in nascent roots and it will be possible to increase the content of these two active compounds through transgenic technology.

## 3. Discussion

Expanding the scale of *Taxus* planting through cutting propagation stands out as a primary solution to address the shortage of paclitaxel. Traditional cutting techniques typically involve using woody adventitious buds (with a branch diameter of 0.2–0.3 cm) from 1- to 4-year-old branches for propagation. The propagation cycle usually involves cutting in December, rooting around May of the following year, and taking approximately 6–8 months for the roots to reach a length of 5–8 cm. The slow growth of the cutting roots greatly limits the development of the *Taxus* industry. In this study, the branches of *T. media* aged 1–3 years were selected for root induction treatment using *A. rhizogenes*. The number of roots and rooting rates in *T. media* branches treated with *A. rhizogenes* was significantly higher than those without this treatment. The transplanted branches with induced roots showed high survival rates, fast growth, and robust development. The use of *A. rhizogenes*-induced hairy root transformation techniques has been reported in various species, including cotton [8], soybean [5], lychee [9], fig [10], pea [11], and *Taxus* [11,12]. However, most of these studies focused on inducing hairy root formation in stem segments or leaf tissues through genetic engineering to increase secondary metabolite content or perform gene editing. The reported applications of *A. rhizogenes* in *T. media* mainly involved genetic transformation of detached hairy roots or cell suspension cultures [12,13] and not root induction in living plants as demonstrated in this study. The Ri plasmid of *A. rhizogenes* contains the transfer DNA (T-DNA) region associated with hairy root formation (known as TL-DNA and TR-DNA) and the virulence region (Vir region). The TL-DNA region contains four rol genes (*rolA*, *rolB*, *rolC*, and *rolD*) related to hairy root morphogenesis, while the TR-DNA region contains the *aux* gene involved in the synthesis of plant growth hormones and the *ags* gene encoding the enzyme responsible for cytokinin biosynthesis [4]. Although it has been reported that the main reason *A. rhizogenes* promotes hairy root formation is due to the Ri plasmid’s ability to induce hairy root formation in plants, the molecular mechanisms underlying this induction are not fully understood. In this study, transcriptomic and metabolomic analyses revealed significant changes in amino acid metabolism, nucleic acid metabolism, plant hormone signaling pathways, secondary metabolite production, and other processes closely related to rooting rates, the number of adventitious roots, and growth and development in roots induced by *A. rhizogenes.* It is feasible to enhance paclitaxel content through transgenic approaches mediated by rooting *A. rhizogenes*.

To date, hairy root cultures have been obtained from more than 100 plant species, including several endangered medicinal plants, and have acted as bioreactors applied to improve the yields of desired metabolites and to produce recombinant proteins [14]. As reviewed by Srivastava and Srivastava [15], these genetically transformed root cultures can produce levels of secondary metabolites comparable to that of intact plants, such as flavors, terpenoids, and so on. As showed in this research, the secondary metabolites were rich in TR and the anticancer ingredients taxol and baccatin III were also detected in the TR, of which the content was far more than 1/20 of the 20-year-old root. Moreover, phytohormones including (+)-abscisic acid, Indole-3-carboxaldehyde, and Zeatin were detected in the TR and WR; the Zeatin was most enriched in the TR, providing a theoretical basis for the growth of hairy roots on hormone-free medium and a bioreactor for value natural metabolites [15].

## 4. Materials and Methods

### 4.1. Plant and Substrate Materials

For rooting, 1–3-year-old branches (with a stem diameter of 0.8–1.2 cm) of healthy and vigorous *T. media* trees were selected. Before treatment, the needles and buds were removed from the branches. A ring about 5 mm wide was formed on each branch by removing the bark and bundle cambium layer using a sterilized ring cutter. It was important to remove the cambium layer cleanly at the girdling site to prevent wound healing and root formation. The girdling area should be immediately wrapped and kept moist.

The substrate formula used in this experiment was prepared in a ratio of native soil (representing the local cultivated soil): perlite: nutrient soil: vermiculite: nutrient solution = 4:1:4:2:1. The nutrient solution included 0.02 wt% indole-3-acetic acid (IAA, purity 100%), 0.01 wt% naphthyl acetic acid (NAA, purity 100%), 0.13 wt% potassium permanganate (purity 99%), and 2 wt% plant growth regulator (standard: NY1428-2007 [16]). The pH of the substrate ranged from 6.5 to 7.5. The moisture content of the substrate was maintained between 55% and 75%. If the humidity did not meet the standard after adding the nutrient mixture, it was adjusted by spraying a small amount of water multiple times.

### 4.2. A. rhizogenes Strains

*A. rhizogenes* strains MSU440, Ar1193, C58C1, ArA4, ArQual, and ATCC15834 (Shanghai Weidi Biotechnology Company, Shanghai, China).

### 4.3. A. rhizogenes-Mediated Branch Rooting Transformation System

*A. rhizogenes* strains were activated with an LB solid medium and the mono-clone was transferred to a new liquid LB medium. The mixture was subsequently incubated at 28 °C with shaking at 180 rpm until the value of OD600 = 0.5 − 0.6. The cultures were then centrifuged for 10 min at 4800× *g* and the pellets were resuspended in 1/2MS liquid medium containing 200 μM acetosyringone (AS) to the desired volume. The suspensions were induced at 28 °C with shaking at 180 rpm for 3 h and were then sprayed on the girdling sites, especially focused on the upper girdling wound as it was crucial for root formation. The plant high-pressure propagation box or traditional film bundle with substrate is fixed at the girdling site, which must be located in the middle of the box and cannot be moved. Before fixing the high-pressure propagation box, the substrate surface is sprayed with *A. rhizogenes* to better bind to the wound site and promote rooting. Then, the new shoots on the treated branch were removed to reduce nutrient loss and a tag was hung and used as a rope to support the branch in an upright position. At last, regularly inspect the moisture level of the soil at the wrapping site to ensure proper humidity. The experimental design included 1 blank control group (no treatment with *A. rhizogenes*) and 6 treatment groups (6 different treatments with *A. rhizogenes*), with 3 biological replicates per experiment. Each replicate contains at least 35 branch rooting treatments.

### 4.4. Sample Preparation for Multi-Omics Analysis

After induction treatment for about 7 months, the adventitious roots induced by *A. rhizogenes* MSU440 (TR) and those formed naturally without *A. rhizogenes* treatment (WR) were collected and washed with sterile PBS and frozen in liquid N_2_. Collected samples were used for further transcriptome and metabolome analysis. Each sample had three repetitions.

### 4.5. Metabolites Extraction

The freeze-dried samples were crushed with a mixer mill for 60 s at 60 Hz. Then, a 50 mg aliquot of individual samples was precisely weighed and transferred to an Eppendorf tube after the addition of 700 μL of extract solution (methanol/water = 3:1, precooled at −40 °C, containing internal standard). After a 30 s vortex, the samples were homogenized at 35 Hz for 4 min and sonicated for 5 min in an ice-water bath. Next were then repeat homogenized and sonicated 3 times. Then, the samples were extracted overnight at 4 °C on a shaker and centrifuged at 12,000 rpm (RCF = 13,800× *g*, R = 8.6 cm) for 15 min at 4 °C. The supernatant was carefully filtered through a 0.22 μm microporous membrane; then, the resulting supernatants were diluted 20 times with a methanol/water mixture (*v*:*v* = 3:1, containing internal standard), vortexed for 30 s, and transferred to 2 mL glass vials. Subsequently, 100 μL from each sample was taken and pooled as QC samples. They were then stored at −80 °C until UHPLC-MS analysis (Allwegene Technology Co., Ltd., Beijing, China).

### 4.6. UHPLC-MS-MS Analysis

Ultra-high-performance liquid chromatography (UPLC) chromatographic separation was performed using an EXIONLC System (Sciex) installed with a Waters UPLC column (Waters Acquity UPLC HSS T3 1.8 μm 2.1 × 100 mm) [17]. Mobile phase A consisted of 0.1% formic acid (FA) in H_2_O and mobile phase B consisted of 100% acetonitrile (ACN). The following gradient program was used: 98% B, 0.5 min; 98–50% B, 0.5–10 min; 50–5% B, 10–11 min; 5% B, 13 min; 5–98% B, 13–13.1 min; and 95% B, 15 min. The flow rate was set at 0.5 mL/min. The injection volume was 2 µL and the samples were maintained at 4 °C in the autosampler. A Sciex QTrap 6500+ (Sciex, Foster City, CA, USA) was applied for assay development. Typical ion source parameters were IonSpray voltage: +5500/−4500 V, curtain gas: 35 psi, temperature: 400 °C, ion source gas 1:60 psi, ion source gas 2:60 psi, and DP: ±100 V. 

### 4.7. Metabolome Analysis

SCIEX Analyst Work Station Software (Version 1.6.3) was employed for MRM data acquisition and processing. MS raw data (.wiff) files were converted to the TXT format using an MSconventer. An in-house R program and database were applied to peak detection and annotation.

X peaks were detected and X metabolites could be left through the following process. First of all, the metabolite feature is detected in <20% of experimental samples or detected in <50% of quality control (QC) samples and it is removed from data analysis. Then, the missing values of raw data were filled up by half of the minimum value. In addition, the internal standard normalization method was employed in this data analysis. Finally, features with relative standard deviation (RSD) > 30% should be removed from the subsequent analysis. The resulting three-dimensional data involving the peak number, sample name, and normalized peak area were fed to the R package metaX for principal component analysis (PCA) and orthogonal partial least squares discriminant analysis (OPLS-DA) [18]. Principal component analysis (PCA) showed the distribution of origin data. In order to obtain a higher level of group separation and obtain a better understanding of variables responsible for classification, supervised orthogonal partial least squares discriminant analysis (OPLS-DA) was applied. Then, a 7-fold cross-validation was performed to calculate the value of R2 and Q2. R2 indicates how well the variation of a variable is explained and Q2 means how well a variable can be predicted. Afterward, to check the robustness and predictive ability of the OPLS-DA model, a 200 times permutation was further conducted. Afterward, the R2 and Q2 intercept values were obtained. Here, the intercept value of Q2 represents the robustness of the model, the risk of overfitting, and the reliability of the model, which will be the smaller the better. To refine this analysis, the first principal component of variable importance in the projection (VIP) was obtained. The VIP values summarize the contribution of each variable to the model. The metabolites with VIP > 1 and *p* < 0.05 (student *t*-test) were considered as significantly changed metabolites (differentially expressed metabolites, DEMs). In addition, commercial databases including MetaboAnalyst 5.0 http://www.metaboanalyst.ca/ (accessed on 29 June 2023) and KEGG http://www.kegg.jp (accessed on 20 June 2023) were utilized to search for the pathways of metabolites.

### 4.8. RNA-Seq

Total RNA was extracted using the TRIzol method (Invitrogen, Waltham, CA, USA) and treated with RNase-free DNase I (Takara, Kusatsu, Japan). A total amount of 1.5 μg RNA per sample was used as input material for the RNA sample preparations. Sequencing libraries were generated using NEBNext^®^ Ultra™ RNA Library Prep Kit for Illumina^®^ (NEB, Ipswich, MA, USA) following manufacturers’ recommendations and index codes were added to attribute sequences to each sample. The double-strand cDNA segments with inserts of 200–250 bp sequence tags were purified using the AMPure XP system (Beckman Coulter, Beverly, MA, USA). A cDNA library was constructed by PCR amplification, the library preparations were sequenced on an Illumina Novaseq 6000 platform by the Beijing Allwegene Technology Company Limited (Beijing, China), and paired-end 150 bp reads were generated. All clean tags were mapped to the NCBI database and the genome of *T. chinensis* [19].

Differential expression analysis of the two groups was performed using the DESeq2 R package (1.16.1). DESeq2 provides statistical routines for determining differential expression in digital gene expression data using a model based on the negative binomial distribution. The resulting *p*-values were adjusted using the Benjamini and Hochbergs approach for controlling the false discovery rate. Genes with an adjusted *p*-value < 0.05 found by DESeq2 and absolute |Log_2_ (Fold change)| > 0 were assigned as the threshold for significantly differential expression. GO enrichment analysis of differentially expressed genes was implemented by the clusterProfiler R package, in which gene length bias was corrected. The cluster profiler R package was used to test the statistical enrichment of differential expression genes in KEGG pathways.

### 4.9. LC-MS-Targeted Quantitative Detection of Paclitaxel and Baccatin III

In total, 200 mg of TR, WR, and 20-year-old roots were weighed and placed in 2 mL centrifuge tubes, respectively. Overall, 600 µL of 80% methanol was added to the tube. The mixture was sonicated at 4 °C for 30 min and kept at 4 °C for 1 h and then was centrifuged at 4 °C, 12,000 rpm for 15 min. The supernatant was collected and diluted 100-fold for LC-MS/MS analysis. The chromatographic column was Acquity UPLC HSS T3 (1.8 µm, 2.1 mm × 100 mm). The UPLC-QTrap-MS separation conditions were as follows: column temperature: 40 °C; flow rate: 0.30 mL/min; mobile phase composition: A-water (0.1% formic acid), B-acetonitrile; total runtime: 5 min; and injection volume: 6 µL. The standard solution was prepared according to the chromatographic and massed spectrometry conditions mentioned above and then was transferred into the sample vial for injection. An appropriate amount of the standard solution was taken accurately and diluted with 80% methanol to create a suitable series of standard solutions. The standard curve was plotted based on the peak areas of the standard samples at different concentrations. The contents of paclitaxel and baccatin III were calculated using the standard curve and MultiQuant software version 3.0.

## 5. Conclusions

The *Taxus* genus is a vital source of the anticancer drug paclitaxel, which faces supply–demand challenges. Expanding *Taxus* planting through traditional cutting propagation is slow and inefficient, hindering industry growth and paclitaxel utilization. Establishing efficient stem rooting systems is essential for *Taxus* propagation and root utilization. This study used *T. media* branches, rich in paclitaxel, induced rooting via *A. rhizogenes*. All strains improved rooting, with MSU440 and C58 being more effective. The induced roots absorbed nutrients had a 100% transplant survival rate and influenced hormone signaling, amino acid metabolism, zeatin synthesis, and secondary metabolites via transcriptome and metabolome analysis. Paclitaxel content remained consistent in induced and natural roots. These findings support *T. media* propagation and *Taxus* industry sustainability.

## Figures and Tables

**Figure 1 ijms-25-00375-f001:**
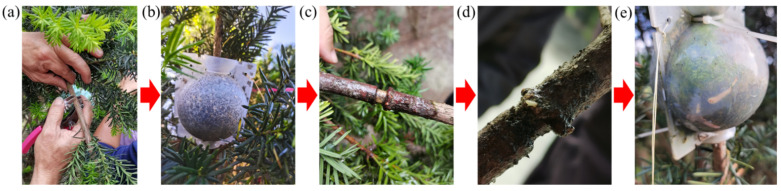
The technical process of branch rooting of *T. media*. (**a**) Girdling treatment of 1–3-year-old branches. (**b**) Plant high-pressure propagation box fixing at the girdling site. (**c**) Swelling and formation of root apical meristem at the upper girdling site. (**d**) Formation of young roots at the upper girdling site. (**e**) Elongation and growth of branch adventitious roots.

**Figure 2 ijms-25-00375-f002:**
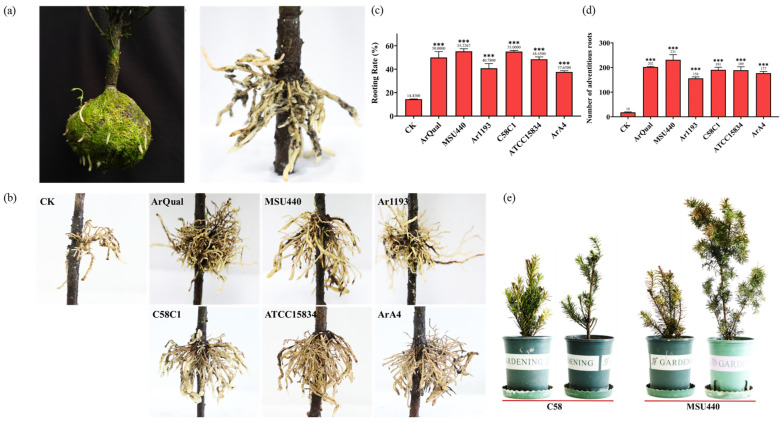
Demonstration of branch rooting. (**a**) Morphology of roots after opening the high-pressure propagation box, with new shoots emerging from the branch. (**b**) The morphology of roots formed under the treatment of different *A. rhizogenes* strains, showing their growth from the girdling site. CK means the normally formed roots of *T. media* branches without the treatment of *A. rhizogenes*. (**c**) The rooting rate of *T. media* branches under the treatment of different *A. rhizogenes* strains that, without the treatment of *A. rhizogenes,* was as a control (CK). (**d**) The number of adventitious roots in *T. media* branches under treatments of different *A. rhizogenes* strains that, without the treatment of *A. rhizogenes,* was as a control (CK). (**e**) The growth status of the rooting branches induced by *A. rhizogenes* strains MSU440 and C58 after being transplanted into pots. Each experiment was repeated three times. Each replicate contains at least 35 branch rooting treatments. Asterisks indicate significant differences (*** *p* < 0.001).

**Figure 3 ijms-25-00375-f003:**
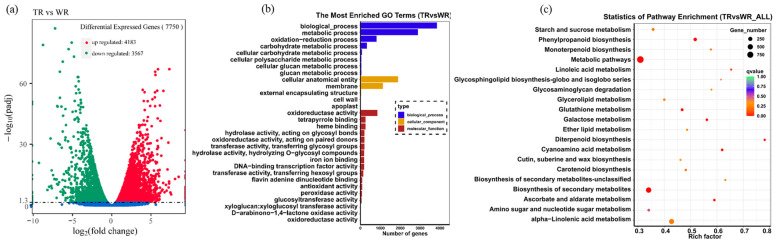
Analysis of differentially expressed genes (DEGs) in TR vs. WR; gene ontology (GO); and the Kyoto Encyclopedia of Genes and Genomes (KEGG) enrichment analysis of DEGs in TR vs. WR. (**a**) Differentially expressed genes in TR vs. WR. (**b**) GO enrichment analysis showed that biological processes, cellular anatomical entity, and oxidoreductase activity were involved in the biological process, cellular component, and molecular function, respectively. (**c**) KEGG enrichment analysis showed that metabolic pathways, biosynthesis of secondary metabolites, and phenylpropanoid biosynthesis were the top three significant enrichment pathways.

**Figure 4 ijms-25-00375-f004:**
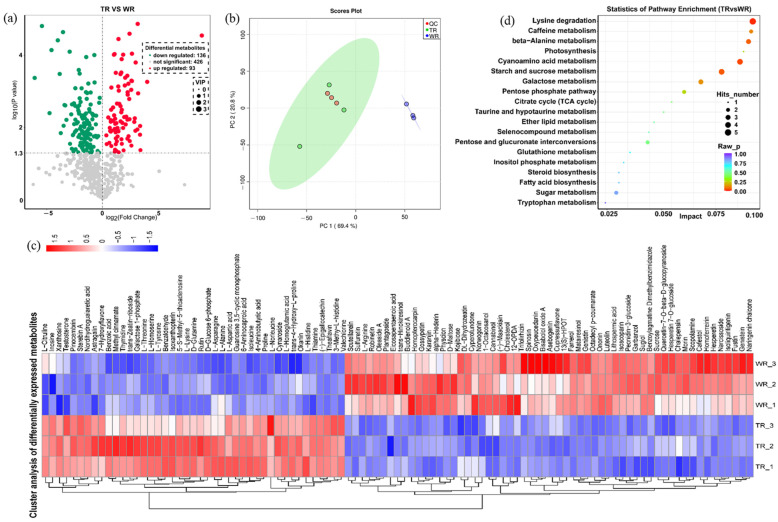
Analysis of differentially expressed metabolites (DEMs) in TR vs. WR. (**a**) Differentially expressed metabolites in TR vs. WR. (**b**) Principal component analysis revealed clear differences between the metabolites in TR and WR. (**c**) Cluster analysis of DEMs. (**d**) KEGG enrichment analysis showed that DEMs were the main components for the biosynthesis of amino acids and their derivatives.

**Figure 5 ijms-25-00375-f005:**
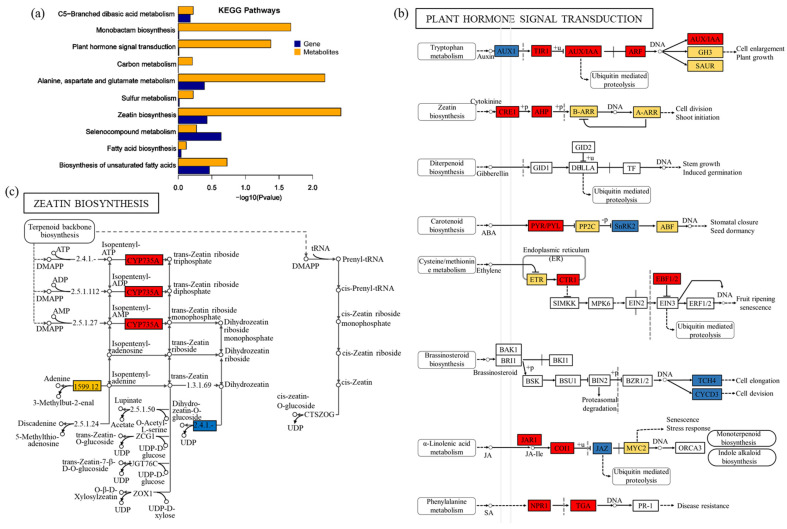
Combined analysis of transcriptome and metabolome (DEGs/DEMs) in TR vs. WR. (**a**) Top ten KEGG pathways correlated with DEGs/DEMs in TR vs. WR. (**b**) Plant hormone signal transduction pathway in the correlated DEGs/DEMs in TR vs. WR. (**c**) Zeatin biosynthesis pathway in the correlated DEGs/DEMs in TR vs. WR. In (**b**,**c**), the red rectangles indicate the DEGs encoded in the protein were all upregulated, the blue rectangles indicate the DEGs encoded in the protein were all downregulated and the yellow rectangle indicates the genes encoded in the protein contain both upregulated and downregulated DEGs.

**Figure 6 ijms-25-00375-f006:**
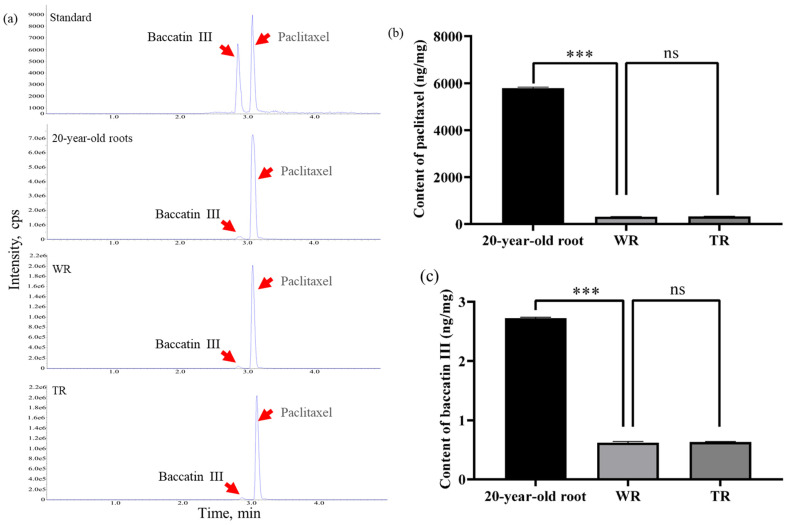
The quantitative detection of paclitaxel and baccatin III. (**a**) The LC-MS/MS analysis of paclitaxel and baccatin III in 20-year-old roots, WR, and TR. The retention time (RT) of paclitaxel is 3.08 min and that of baccatin III is 2.86 min. (**b**) The content of paclitaxel in 20-year-old roots, WR, and TR. (**c**) The content of baccatin III in 20-year-old roots, WR, and TR. Asterisks indicate significant differences (*** *p* < 0.001).

## Data Availability

The data and materials supporting the conclusions are contained. within the article and the Appendix A. The raw sequence data are available at the NCBI (https://www.ncbi.nlm.nih.gov/bioproject/PRJNA999506/, accessed on 20 June 2023) under the accession number SRA: PRJNA999506 on 28 July 2023.

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
