# Peer review of "Technology Invention and Mechanism Analysis of Rapid Rooting of *Taxus × media* Rehder Branches Induced by *Agrobacterium rhizogenes"

_ijms, 2023, doi:10.3390/ijms25010375_

Round 1
Reviewer 1 Report
Comments and Suggestions for Authors
I found this manuscript to be very interesting and it describes an important new method to produce Taxus plants. This could have significant effects on the availability of the drug.
The manuscript is understandable as written but could use some fine edited for additional clarity. The discussion is a little weak and I suggest that some of the results are really discussion -- see manuscript. The font on some of the illustrations needs to be increase as the smaller fonts are difficult to read, especially to older readers. :)
In total, a nice piece of noteworthy work, but needs some close editing and some reorganization.

The manuscript is basically understandable but would be improved by close editing.
Author Response
|
Response to Reviewer 1 Comments
|
||
|
1. Summary |
|
|
|
Thank you very much for taking the time to review this manuscript. Please find the detailed responses below and the corresponding revisions in the re-submitted files.
|
||
|
2. Questions for General Evaluation |
Reviewer’s Evaluation |
Response and Revisions |
|
Does the introduction provide sufficient background and include all relevant references? |
Yes |
|
|
Are all the cited references relevant to the research? |
Yes |
|
|
Is the research design appropriate? |
Yes |
|
|
Are the methods adequately described? |
Yes |
|
|
Are the results clearly presented? |
Yes |
|
|
Are the conclusions supported by the results?
|
Yes |
|
|
3. Point-by-point response to Comments and Suggestions for Authors |
||
|
Comments 1: [Delete and Taxus in italics] |
||
|
Response 1: Thank you for pointing this out. We agree with this comment. Therefore, we have deleted ‘The genus’ and used Taxus in italics in page 1, line 13.
|
||
|
Comments 2: [Delete] |
||
|
Response 2: Thanks for your suggestion. We have deleted ‘famous’ in page 1, line 13..
Comments 3: [always use italics for Taxus] Response 3: Thanks for your suggestion. We have used Taxus in italics in page 1, line 15, 16, 18, 19, 31, 35, and 37; page 2, line 50, 51, 53, 55, 58, 62, and 74; page 9, line 272, 278, and 284; page 12, line 442, 443; page 13, line 445, 451.
Comments 4: [This sentence needs to be rewritten -- perhaps break to two thoughts.] Response 4: Thanks for your suggestion. We have rewritten this sentence in page 1, line 15-18. [However, traditional cutting propagation of Taxus is difficult to root and time-consuming. And obtaining the roots with high paclitaxel content will cause tree death and resource destruction, which is not conducive to the development of the Taxus industry.]
Comments 5: [delete it.] Response 5: Thanks for your suggestion. We have deleted ‘it’ in page 1, line 19.
Comments 6: [Maybe move A. r. to beginning of sentence] Response 6: Thanks for your suggestion. We have revised the sentence and moved ‘Agrobacterium rhizogenes’ to beginning of sentence in page 1, line 20-22. [Agrobacterium rhizogenes were induced in the 1-3-year-old branches of Taxus × media Rehder, which has the highest paclitaxel content.]
Comments 7: [Delete] Comments 8: [Delete]
Comments 9: [Delete]
Comments 10: [Delete research] Comments 11: [delete second and third cancer] Comments 12: [New paragraph] Comments 13: [Delete]
Comments 14: [use whereas as comparison]
Comments 15: [move to after dicot. . . eliminate second plants]
Comments 16: [seven]
Comments 17: [All transplanted branches survived]
Comments 18: [Not heard this term before]
Comments 19: [Lettering in b and c needs to be made larger. Difficult to read as is]
Comments 20: [do not use capital letters for PGRs] Response 20: Thanks for your suggestion. We have revised the capital letter in page 5, line 162
Comments 21: [Increase font of lettering for figures.] Response 21: Thanks for your suggestion. We have increased the font of lettering for figures in page 6.
Comments 22: [This seems to me to be discussion. Should not cite literature in your results.] Response 22: Thanks for your suggestion. We have revised this paragraph in page 6, line 211-223. [A crucial aspect of plant hormone function involves the regulation of diverse developmental processes. Hormones such as auxin, jasmonic acid (JA), gibberellic acid (GA), cytokinin, abscisic acid (ABA), ethylene, brassinosteroid, salicylic acid (SA), and strigolactone (SL) employ mechanisms that activate transcriptional regulators for degradation. As shown in Figure 5b and Table S7, in the auxin signaling pathway, DEGs encoding Aux1 were consistently downregulated, while those for TIR1, AUX/IAA, and ARF were consistently upregulated. Cytokinin, known for its role in cell proliferation, exhibited upregulation of core DEGs, including CRE1 and AHP. ABA, a critical endogenous messenger in plant stress responses, showed upregulation of DEGs encoding ABA receptor genes PYR/PYL, but downregulation of those encoding the pivotal regulator SnRK2. In the ethylene signaling pathway, DEGs encoding protein kinase CTR1 and F-Box proteins EBF1/2 were upregulated. Moreover, genes (JAR1, COI1, NPR1, and TGA) involved in JA and SA signal transduction pathways were upregulated.]
Comments 23: [Seems to repeat the introduction] Response 23: Thanks for your suggestion. We have revised this paragraph in page 9, line 271-272. [Expanding the scale of Taxus planting through cutting propagation stands out as a primary solution to address the shortage of paclitaxel.]
Comments 24: [Should be numbers here and in text? Most mdpi journals use this format. Please check.] Response 24: Thanks for your suggestion. We have cited all the references in numbers. |
||

Reviewer 2 Report
Comments and Suggestions for Authors
Dear authors,
This is an important study on the formation of adventitious roots on branches of Taxusxmedia induced by Agrobacterium rhyzogenes. The authors showed the improvement of adventitious root formation by using different strains of A. rhyzogenes. The manuscript integrated transcriptomics and metabolomics analyses to study the effect of A. rhizogenes on the hormone signal transduction, amino acid metabolism, or secondary metabolite synthesis pathway in roots of Taxus × media.
In my opinion the topic is relevant and the manuscript is appropriate and worth to be published in the "International Journal of Molecular Sciences".
However, it presents some shortcomings that need to be addressed before publication
My comments:
1-In the figure 2, is shown the morfology of root system of WR and TR. It seems that TR have more secondary roots than WR . This feature was not mentioned or discussed. Root system architecture triggered by Agrobacterium is different to the CK root system. The number of roots as well as the length of roots developed in Agrobacterium-treated and control branches should be included in the results section.
2-WR and TR have been used in the metabolomic and transcriptomic anlyses, but it is not indicated whether the whole roots or specific areas of the roots (apical, division , elongation or maturation zones) were used. It should be defined. For instance, auxin concentration or signalling pathways can vary along the root. This point could be relevant for quantitative detection of paclitaxel and baccatin III in WT, TR and roots from a 20-year old roots.
2-In the figure 2, is shown the morfology of root system in in WR and TR. It seems that TR roots have more msecondary roots than WR . This feature was not mentioned or discussed.
3-The discussion must be improved. The results of the transcriptomic and metabolomic analyses arepoorly discussed.
4- Some abbreviations first appear in the text are not use the full name, such as FA, QC
Author Response
|
Response to Reviewer 2 Comments
|
||
|
1. Summary |
|
|
|
Thank you very much for taking the time to review this manuscript. Please find the detailed responses below and the corresponding revisions in the re-submitted files.
|
||
|
2. Questions for General Evaluation |
Reviewer’s Evaluation |
Response and Revisions |
|
Does the introduction provide sufficient background and include all relevant references? |
Yes |
|
|
Are all the cited references relevant to the research? |
Yes |
|
|
Is the research design appropriate? |
Yes |
|
|
Are the methods adequately described? |
Can be improved |
|
|
Are the results clearly presented? |
Yes |
|
|
Are the conclusions supported by the results?
|
Can be improved |
|
|
3. Point-by-point response to Comments and Suggestions for Authors |
||
|
Comments 1: [In the figure 2, is shown the morfology of root system of WR and TR. It seems that TR have more secondary roots than WR . This feature was not mentioned or discussed. Root system architecture triggered by Agrobacterium is different to the CK root system. The number of roots as well as the length of roots developed in Agrobacterium-treated and control branches should be included in the results section.] |
||
|
Response 1: Thank you for pointing this out. We agree with this comment. We have recorded the adventitious roots in Taxus media branches under different treatments, with 10 branches taken for each treatment. Results showed that untreated branches produced fewer adventitious roots, only 40±13 (n=10) in total; branches treated with A. rhizogenes generated a higher number of adventitious roots, with the MSU440 treatment showing the most significant effect. In the manuscript, we have included an image (Figure 1d) along with the following content in page 2, line 85-88. Meanwhile, untreated branches produced fewer adventitious roots, only 40±13 (n=10) in total. [In contrast, branches treated with A. rhizogenes generated a higher number of adventitious roots, with the MSU440 treatment showing the most significant effect (Figure 2d)].
|
||
|
Comments 2: [WR and TR have been used in the metabolomic and transcriptomic anlyses, but it is not indicated whether the whole roots or specific areas of the roots (apical, division , elongation or maturation zones) were used. It should be defined. For instance, auxin concentration or signalling pathways can vary along the root. This point could be relevant for quantitative detection of paclitaxel and baccatin III in WT, TR and roots from a 20-year old roots.] |
||
|
Response 2: Thanks for your suggestion. We did not separately analyze root tips, root caps, etc.; instead, we used all the adventitious roots that had grown. Our objectives were two fold: firstly, for transplantation to shorten the breeding cycle, and secondly, to extract paclitaxel content from the entire living adventitious roots and the suspended cell line of adventitious roots for broader applications in production. Therefore, we did not further focus on specific parts of the roots. However, we will pay attention to the changes and functions of each part of the adventitious roots in subsequent research.
Comments 3: [In the figure 2, is shown the morfology of root system in in WR and TR. It seems that TR roots have more msecondary roots than WR . This feature was not mentioned or discussed.] Response 3: Thank you for pointing this out. We agree with this comment. Based on your suggestion, we carefully observed and recorded the number of adventitious roots under different treatments. To our delight, the number of roots in T. media branches treated with A. rhizogenes was significantly higher than those without this treatment. It is possible that the A. rhizogenes contains genes related to the synthesis of inducing hormones, thereby inducing the formation of adventitious roots. Moreover, during the induction of root formation, the A. rhizogenes also induced the synthesis of secondary metabolites, providing a theoretical basis for the extraction of important secondary metabolites through adventitious root. Additionally, it offers feasibility for enhancing paclitaxel content through transgenic approaches mediated by rooting A. rhizogenes. We included the following content in page 8, line 275-276 [The number of roots and rooting rates in T. media branches treated with A. rhizogenes was significantly higher than those without this treatment.] and in page 9, line 293-298 [In this study, transcriptomic and metabolomic analyses revealed significant changes in amino acid metabolism, nucleic acid metabolism, plant hormone signaling pathways, secondary metabolite production, and other processes closely related to rooting rates, the number of adventitious roots, growth and development in roots induced by A. rhizogenes. It is feasible to enhance paclitaxel content through transgenic approaches mediated by rooting A. rhizogenes.].
Comments 4: [The discussion must be improved. The results of the transcriptomic and metabolomic analyses are poorly discussed.] Response 4: Thanks for your suggestion. We have improved the analysis of the omics data, but the focus is still on A. rhizogenes promoting rooting and metabolite accumulation. A point that there are more secondary metabolites in TR was added in the results and discussion part in page 9, line 299-310 [To date, hairy root cultures have been obtained from more than 100 plant species, including several endangered medicinal plants, and be as a bioreactor applied to improve the yields of desired metabolites and to produce recombinant proteins. As reviewed by Srivastava and Srivastava, these genetically transformed root cultures can produce levels of secondary metabolites comparable to that of intact plants, such as flavors, terpenoids, and so on. As showed in this research, the secondary metabolites were rich in TR, and the anticancer ingredients-taxol and baccatin III were also detected in the TR, of which content were far more than 1/20 of 20-year-old root. Moreover, phytohormones including (+)-abscisic acid, Indole-3-carboxaldehyde, Zeatin were detected in the TR and WR, and the Zeatin was most enriched in the TR, providing a theoretical basis for the growth of hairy roots on hormone-free medium and as bioreactor for value natural metabolites.].
Comments 5: [Some abbreviations first appear in the text are not use the full name, such as FA, QC] Response 5: Thanks for your suggestion. We have used full names for abbreviations such as DEG, KEGG, FA, ACN, QC, and RSD etc. the first time they appear in page 4, line 125, 135; page 10, line 374, 375; page 11, line 389, 392. |
||

Round 2
Reviewer 2 Report
Comments and Suggestions for Authors
Dear Authors,
All my suggestions have been addressed in the new version of the manuscript
I do not have further comments.